# Herbivores rescue diversity in warming tundra by modulating trait-dependent species losses and gains

Elina Kaarlejärvi [1,2], Anu Eskelinen[3,4,5] & Johan Olofsson[1]

Climate warming is altering the diversity of plant communities but it remains unknown which species will be lost or gained under warming, especially considering interactions with other factors such as herbivory and nutrient availability. Here, we experimentally test effects of warming, mammalian herbivory and fertilization on tundra species richness and investigate how plant functional traits affect losses and gains. We show that herbivory reverses the impact of warming on diversity: in the presence of herbivores warming increases species richness through higher species gains and lower losses, while in the absence of herbivores warming causes higher species losses and thus decreases species richness. Herbivores promote gains of short-statured species under warming, while herbivore removal and fertilization increase losses of short-statured and resource-conservative species through light limitation. Our results demonstrate that both rarity and traits forecast species losses and gains, and mammalian herbivores are essential for preventing trait-dependent extinctions and mitigate diversity loss under warming and eutrophication.

[1] Climate Impacts Research Centre (CIRC), Department of Ecology and Environmental Science, Umeå University, SE-981 07 Abisko, Sweden. [2] Department of Biology, Vrije Universiteit Brussel (VUB), Pleinlaan 2, 1050 Brussels, Belgium. [3] Department of Physiological Diversity, Helmholtz Center for Environmental Research—UFZ, Permoserstr. 15, D-04318 Leipzig, Germany. [4] German Centre for Integrative Biodiversity Research (iDiv) Halle-Jena-Leipzig, Deutscher Platz 5e, D-04103 Leipzig, Germany. [5] Department of Ecology, University of Oulu, P.O. Box 3000, FI-90014 Oulu, Finland. Correspondence and requests for materials should be addressed to E.K. (email: elina.kaarlejarvi@umu.se)

Global climate warming is dramatically altering composition of communities, but its effects on local biodiversity have proven difficult to predict[1, 2]. Local diversity is formed by two opposite processes, species losses and gains (i.e., extinctions and immigrations), and the net effect of warming on diversity can be either positive or negative depending on the balance between species gains and losses. Both of these processes are likely to be impacted by warming[3, 4], but are rarely studied simultaneously[5]: Some studies have linked recent climate warming to local species gains as a consequence of immigration and range expansions of warm-adapted species eg. refs [6–8], while others have found evidence for species losses as a result of climate-driven local extinctions or range contractions[4, 8, 9]. Climate-driven losses and gains may contribute to high species turnover in local communities world-wide[10] without altering species richness if gains equal losses, but with pronounced consequences for ecosystem functioning. Therefore, to explain the highly variable changes in local diversity[1, 10, 11] and to foster rigorous forecasts of warming effects on ecosystem functioning, we need mechanistic understanding of how and in which circumstances warming causes species losses and gains.

Species interactions may offset or magnify effects of climatic changes, and are thus one of the major sources of uncertainty in predicting future diversity changes[12–14]. Few studies have examined how biotic interactions interact with climate to contribute to diversity changes and even fewer have simultaneously studied species losses and gains to understand the relative roles of these two opposite forces. For example, herbivores and/or competition with resident species can hamper invasion of novel species[13, 15–17], but we lack knowledge on whether these interactions simultaneously induce range contractions and losses of some other species. This knowledge is essential not only for predicting changes in diversity but also for gaining deeper understanding on how communities and ecosystems change under global changes, and how these changes reflect to human well-being[18].

Species' sensitivity and tolerance to abiotic and biotic changes are determined by their functional traits. By partitioning changes in community diversity to species-level losses and gains we can explore the importance of traits for predicting the effects of climatic changes on local diversity, and identify traits that favor species' immigration or make species susceptible to local extinctions. Generally, plant traits associated with high resource-use efficiency in resource-rich conditions ('resource-acquisitive traits'; e.g., tall stature, high specific leaf area (SLA), high nitrogen and phosphorus concentrations[19, 20], and low concentration of secondary defense compounds[21]) have been positively correlated with competitiveness following shifts towards more growth-promoting conditions, for example, increases in winter temperatures[22], or rainfall[23] and nutrient availailability[24]. In such conditions where productivity is high, competition for light may cause local extinctions[25] of species having predominantly the opposite, resource-conservative and slow-growing traits[22, 26]. However, the resource-acquisitive traits may also make species more vulnerable to herbivory due to their taller canopies and higher nutrient concentrations[27], so fast-growing species that strongly respond to shifts towards more benign conditions may become more readily eaten[28]. Thus, when subjected to multiple interacting stressors (e.g., increases in temperature, herbivory and nutrient limitation), traits that confer an advantage in responses to one stressor might limit responses to another stressor. Consequently, fixed species-specific trait combinations may constrain individual species' competitiveness and thus affect coexistence and diversity.

The rate of climate warming is two to three times higher in tundra ecosystems than the global average[29, 30], so knowledge of traits influencing species losses and gains is particularly important for predicting changes in diversity and functions in tundra. Plant growth in these ecosystems is strongly limited by temperature[31, 32], suggesting that climate warming may promote growth of resident tundra plants and enable immigrations of new species from lower altitudes and latitudes[16], thereby potentially increasing diversity. Since plant growth in tundra is also limited by soil nutrients[31], warming effects are likely to be greater when warmer temperature coincides with high soil nutrients, for example, in many alpine tundra areas with high atmospheric nitrogen deposition[33, 34]. Climate warming may also increase nutrient availability by stimulating decomposition of soil organic matter[35]. As predicted by niche dimensionality[36] and multiple resource limitation[37] theories, higher temperature and increased nutrient availability may synergistically increase productivity and their joint effect on diversity may be negative. Warming and nutrient addition together might promote losses of less competitive species and be prerequisites for gains of highly competitive species from lower altitudes and latitudes[16, 38].

Concurrently, tundra ecosystems are also heavily grazed by mammalian herbivores (i.e., reindeer, voles, lemmings)[39], and herbivory can severely limit plant growth and productivity in these systems[40, 41]. Warming impacts on tundra plants can therefore critically depend on herbivory[42–45], in the same way as the effects of nutrient enrichment depend on herbivory[46], and warming may increase richness only under herbivory. However, it remains unclear how warming interacts with herbivory to impact diversity via local immigration and extinction, and which traits predict species' susceptibility to respond to these abiotic and biotic factors. Moreover, although warming effects could depend on both nutrient availability and herbivory, very little is known about three-way interactions of these global change drivers, which in many areas, including tundra, are likely to impact plants simultaneously. Here, we tested the joint effects of warming, mammalian herbivory and fertilization on plant species richness, and partitioned richness changes to species gains and losses in a 5-year field experiment in open tundra in NW Finland. We also evaluated whether functional traits associated with plants' growth potential and sensitivity to herbivory could predict the likelihood of species being lost or gained. Since rare species can have higher extinction risk due to their small population sizes[24, 47, 48], we tested the importance of trait-based functional mechanisms after taking species' initial abundance (a proxy for rarity) into account. We posed the following three hypotheses: (1) Warming should increase species richness in the presence of herbivores by promoting species gains, but lead to increases in biomass and light limitation and hence to greater species losses in the absence of herbivores. (2) Fertilization should amplify the negative impact of warming in the absence of herbivores and further increase species losses and diminish diversity. (3) Resource-conservative traits should increase species' local extinction risks especially in high-biomass and light-limited conditions, while resource-acquisitive traits should confer lower extinction risks in the same conditions.

We show that warming increases tundra diversity in the presence of herbivores by promoting species gains and reducing losses, while in the absence of herbivores warming leads to greater species losses and thus decreases diversity. Herbivores promote diversity by reducing light limitation and diminishing losses of slowly growing short-statured plants. Fertilization amplifies the negative impacts of warming in the absence of herbivores. Our results underscore the critical role of herbivory as a key modulator of impacts of climate warming and nutrient enrichment on trait-dependent species losses and gains, and diversity of tundra plant communities.

## Results

**Herbivores modulate diversity via species losses and gains.** Warming by ~1.5 °C increased total plant species richness in the

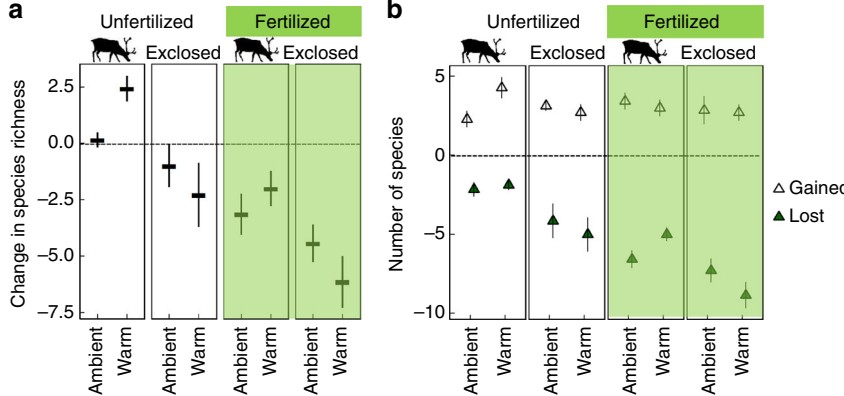

**Fig. 1** Change in plant species richness from 2009 to 2014. **a** Changes in total plant species richness and **b** numbers of plant species (mean ± s.e.m., $n = 7$ for each point) lost and gained under warming, herbivore exclosure and fertilization treatments. Total plant species richness at the start of the experiment was $21.6 ± 0.58$ (mean ± s.e.m.) species in $25 × 50$ cm$^2$ quadrats ($n = 56$)

presence of mammalian herbivores by 2.4 species (13%) on average, but decreased it by 2.3 species (12%) in the absence of herbivores (Fig. 1a, warming (W) × herbivore exclusion (E) interaction, Supplementary Table 1). These changes were a result of both species gains and losses (Fig. 1b), illustrating that both these processes are crucial for understanding factors behind diversity changes. Gains of species (of both vascular plants and bryophytes) were highest and losses smallest in warmed-grazed plots in ambient nutrient availability (Fig. 1b, Supplementary Fig. 1, Supplementary Table 2). In these plots warming likely directly stimulated plant growth, facilitated plant reproductive success and improved survival rates[16, 38], while herbivores reduced community biomass, increased light availability (Fig. 2) and suppressed competitive exclusion, thereby promoting species gains and counteracting losses. These conditions are likely to enhance not only gains of residents (by increasing seed germination and establishment) but also novel species expanding their ranges from lower altitudes and latitudes[16], and thereby increase diversity in tundra. These findings indicate that warmer climate induces species gains (in short time-scale) most strongly in communities with low community biomass and weak light limitation. In contrast, in the absence of herbivores, warming increased species losses, primarily of bryophytes (Supplementary Fig. 1, Supplementary Table 2), indicating that without herbivores warming-induced intensification of light competition can be severe enough to cause local extinctions of bryophytes. This is consistent with expectations, as bryophytes are among the smallest land plants, and thus likely to suffer strongly from competition for light and to respond rapidly to climatic and other environmental changes[49, 50]. Even though warming did not induce losses of vascular plants, the greatest losses of vascular plants were observed in warmed and fertilized plots when herbivores were excluded (five species lost per plot). Since on average one species was lost in plots that were only warmed, these results highlight the synergistic effects of global change, with herbivores playing a particularly important role in influencing diversity.

These changes after 5 years of experimental warming show that local diversity can change rapidly even in perennial tundra ecosystems under climate warming. Strikingly, our results demonstrate that herbivores can mitigate or even reverse negative effects of warming on species richness of whole plant communities and thus serve as an important buffer for maintaining plant diversity under climate warming. This finding suggests that the pan-arctic reduction of species richness observed in warming experiments[51, 52] may only be applicable in ungrazed conditions, and that the presence of native mammalian herbivores may reverse this widely observed climate-diversity relationship.

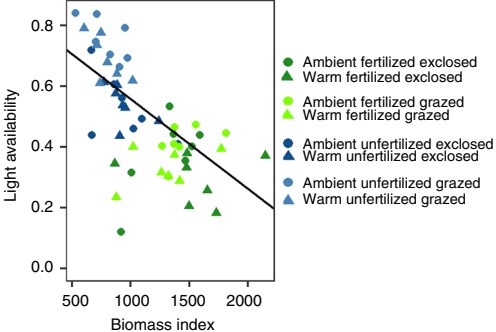

**Fig. 2** Community biomass and light availability. Relationship between total community biomass index in 2014, expressed as hits per 100 pins in $25 × 50$ cm$^2$ quadrats, and light availability measured as proportion of PAR penetrating through vegetation (linear regression, $F_{1,54} = 39.79$, $P < 0.001$, $R^2 = 0.41$, $n = 56$). Warmed plots are indicated by triangles, unwarmed plots (with ambient temperature) by circles. Fertilized plots are shown in *green*, unfertilized in *blue*. Herbivore exclosures are indicated with *dark color*, grazed plots in *light color*

**Importance of soil nutrients**. In accordance with our second hypothesis, fertilization further amplified the reduction of species richness in warmed plots in the absence of herbivores (Fig. 1a), and led to the greatest species losses of both vascular plants and bryophytes (Fig. 1b, Supplementary Table 2, Supplementary Fig. 1). This maximal loss of diversity was associated with the highest community biomass and lowest light availability (Fig. 2), and was mediated solely by species losses, not by lower number of gained species (Fig. 1b, Supplementary Figs. 1, 2). These results demonstrate that conditions leading to high community biomass in tundra increase species losses via light limitation. Our findings also show that by increasing risk of local extinctions, simultaneous alleviation of several growth limiting factors (low temperature and nutrients), although not all resource-based, can lead to loss of diversity[23, 37]. Moreover, warming-induced species gains occurred only in ambient soil nutrient availability, which further suggests that soil nutrient availability is strongly controlling both species losses and gains and shaping the response of tundra diversity under climate warming. Thus, on landscape level, local climate-driven species losses may be strongest in more productive areas with low consumer pressure.

**Rarity and light limitation increase species losses**. Logistic regression modeling revealed that local extinction risk of vascular

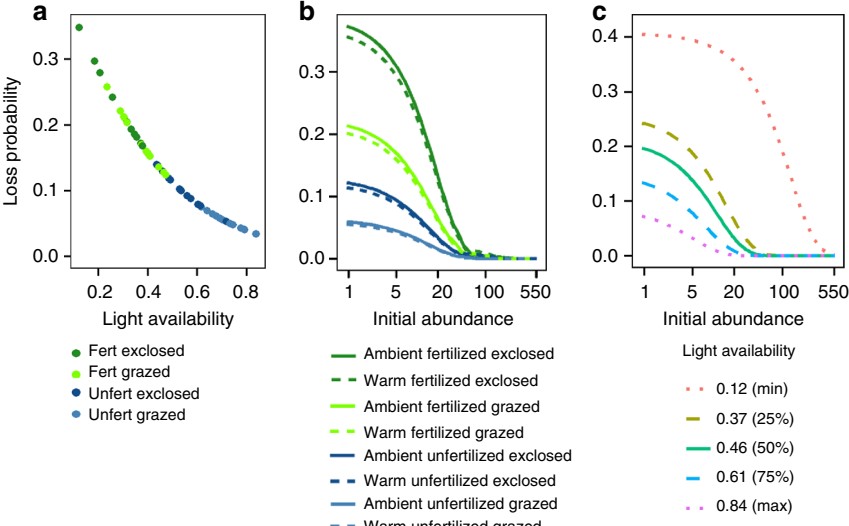

**Fig. 3** Probability of species losses as a function of their initial abundance and light availability. Modeled probability of species losses as functions of **a** light availability (measured as the proportion of ambient PAR), **b** rarity, or species initial abundance (measured as hits per 100 pins per 25×50 cm² quadrat) under warming, grazing and fertilization treatments, and **c** interaction between light availability and initial abundance. In **a** and **b**, fertilized plots are in *green*, unfertilized in *blue*. Herbivore exclosures are indicated with *dark color*, grazed plots in *light color*. In **b**, *dashed line* indicates prediction for warmed plots, *solid line* for unwarmed plots (with ambient temperature). In **c**, *green solid line* shows loss probability when light availability is 50% of ambient PAR, while *dashed lines* indicate 25 and 75% quantiles and *dotted lines* minimum and maximum measured proportion of PAR. Modeling results can be found in Supplementary Table 3

plants increased with diminishing light availability (Fig. 3a, Supplementary Table 3), which was strongly positively correlated with community biomass (Fig. 2). This finding refines earlier indications that light limitation promotes reduction in diversity[46], by showing that it increases probabilities of species losses. Also rarity, measured here as low initial abundance, increased species' probability of being lost, which supports theory[47, 48] and previous findings[24]. However, when light availability declined to <40% of ambient light, loss likelihood of rather abundant species also increased up to four-fold (initial abundance×PAR interaction, Fig. 3b, c, Supplementary Table 3), indicating that not only rare species were lost.

**Trait-dependent species losses**. Species' functional traits influenced their extinction risks, even after accounting for their initial abundance, implying that extinctions were not only random (mainly losses of rare species), but at least partly trait-dependent. Our experimental treatments revealed the importance of some traits for species loss likelihood (Figs. 4 and 5). Although warming clearly contributed to total richness, it did so by influencing gains of all plants and losses of bryophytes, not by causing losses of vascular plants in any treatment combination. Thus, we were unable to identify traits that would predict species loss probability under warming. Since temperature's links with resources are indirect, its association with traits related to resource-use and growth rate (the focal concerns here) may also be predominantly indirect and thus take longer to become evident[22]. Also, some other, more directly temperature-related physiological traits (such as maximal carbon assimilation rate[22] or ability to photosynthesize in low temperature conditions) may also be important for determining the responses of vascular plants to warming. Conversely, both nutrient addition and herbivory induced species losses that were predictively associated with traits. Notably, treatment combinations that increased community biomass and light limitation increased extinction risk of short-statured species; probabilities of their losses were 20% in the absence of herbivores and 55% if soil nutrient limitation was

also alleviated (Fig. 4a). In contrast, increases in community biomass did not increase extinction risk of plants taller than 30 cm (E × fertilization (F) × Height interaction, Fig. 4a, Supplementary Table 3). These findings, together with data from other studies[16, 26, 27], show that plant height is an important determinant of species' competitiveness in resource-rich conditions, and its effects on species' extinction risk are linked to light limitation.

Low relative foliar nitrogen and phosphorus concentrations (high C:N and C:P ratios) increased probabilities of species' losses up to four-fold under the enhanced soil nutrient treatment, with or without herbivory (Fig. 5a, b, Supplementary Table 3). This finding indicates that increases in nutrient availability can be detrimental for species adapted to low nutrient availability, even in the presence of herbivores, and lead to local extinctions. Moreover, we found evidence that a certain set of traits associated with slow growth and high resource investment in tissues makes species vulnerable to any changes that increase community biomass. Low SLA and high concentrations of carbon-based secondary compounds increased species' local extinction risk, especially in high-biomass and low light conditions; and the same set of traits reduced species' likelihood of appearing in a new site, across treatments (Figs. 4b, d, 5c, d, g, h, Supplementary Tables 3, 4). Thus, slowly growing species investing in thick leaves and costly defense compounds warrant particular attention if their populations are small. Collectively, these results confirm that not only abundance-based mechanisms but also trait-based mechanisms influence species' susceptibility to local extinctions[24]. Certain functional traits can thus predispose species to local extinctions under changing biotic and abiotic conditions.

**Trait-dependent species gains**. Although warming, either singly or in combination with other treatments, did not result in species losses, it did increase species gains in unfertilized plots with herbivores. Short-statured species were the most likely to appear in these plots, while the tallest species appeared most often in warmed-fertilized plots that were grazed (W × E × F × Height interaction, Fig. 4c, Supplementary Table 4). Although this trend

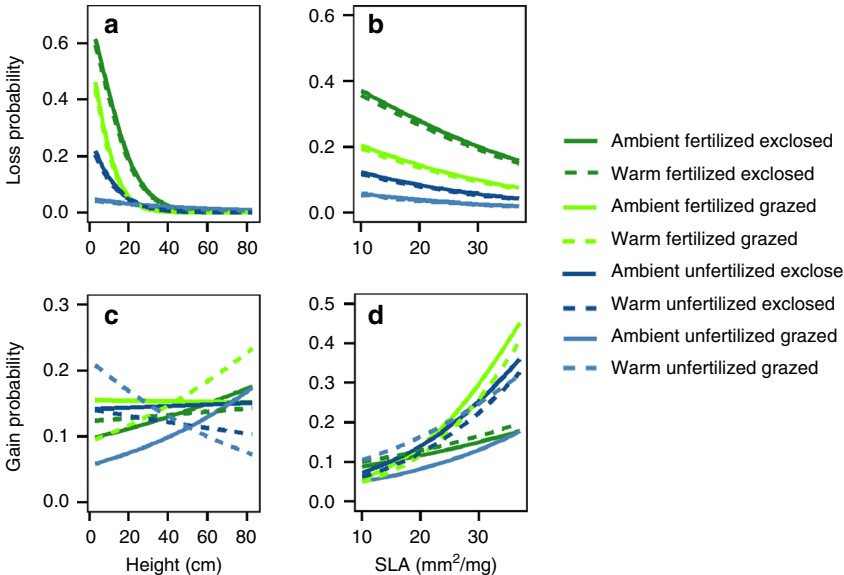

**Fig. 4** Impacts of plant height and specific leaf area on probability of species losses and gains. Modeled probability of species' losses and gains as functions of their height **a**, **c** and SLA **b**, **d** under combinations of warming, grazing and fertilization treatments. Fertilized plots are shown in *green*, unfertilized in *blue*. Herbivore exclosures are indicated with *dark color*, grazed plots in *light color*. *Dashed line* indicates prediction for warmed plots, solid line for unwarmed plots. Modeling results can be found in Supplementary Tables 3, 4)

is not statistically significant (Supplementary Table 4), these results suggest that short-statured species may benefit from warming in grazed habitats, while tall fast-growing species benefit from more benign conditions and increases in available resources, and their appearance in a new site is not initially hampered by herbivores. These findings not only support earlier indications that tall species benefit from warming[16, 22], but also demonstrate that established links between response traits and climatic changes depend on biotic interactions. The likelihood of new species appearing was independent of community biomass (Supplementary Table 4), and species' nutrient ratios were only weakly (C:N) or not at all (C:P) connected to the likelihood of species' gains (Fig. 5e, f, Supplementary Table 4). Overall, species were gained less often than lost, and losses were mainly responsible for changes in diversity, especially under the treatments resulting in the highest biomass and light limitation. The finding that species gains generally played a minor role in diversity responses in our experiment may simply reflect the greater time required for successful immigration and establishment of species in a system. However, once successfully arrived, immigrating species (especially novel competitors of southern/lowland origin) are likely to contribute to diversity change substantially[13, 16].

## Discussion

Our results are experimental evidence from naturally grazed ecosystems confirming the idea, earlier supported by a clipping experiment[53], that grazers can dampen negative effects of warming on diversity. An earlier field study by Post[54] reported no significant effect of grazing or interactive effects of warming and grazing on species richness in 7 years, maybe because it was carried out in species-poor shrub-dominated tundra, whereas ours was done in forb- and grass-dominated tundra with shorter species' life-cycles. Findings from our experiment suggest that local variations in grazing pressure can partly explain inconsistencies in diversity responses to climate change, i.e., why new species appear in some areas, while diversity declines in other areas subjected to similar abiotic changes[1]. However, clearly more work is needed to test generality of our results.

To conclude, we show that herbivores can maintain plant diversity in warming tundra by alleviating light limitation and preventing extinctions of species characterized by short stature and slow growth. These findings highlight the critical role of biotic interactions in modulating effects of climate warming on species losses and gains in local communities, and hence shifts in local diversity. Our results suggest that if climate warming coincides with losses of mammalian herbivores (such as reindeer or lemmings) in tundra, major decreases in diversity are likely as a consequence of losses of short-statured, resource-conservative species. Our results also reveal that such species losses are likely to be even greater in more productive areas, or where N-deposition is high. These findings therefore call for conservation of native mammalian herbivores when attempting to mitigate climate warming-induced local extinctions. Moreover, since the traits related to slow-growth, that increased extinction risk of initially abundant species in our study, are also likely to be associated with ecosystem processes (effect traits)[55], the observed reductions in diversity may have far-reaching consequences for functioning of the ecosystem. The impacts of such trait-dependent losses are likely to be much greater than predicted by numerous biodiversity-ecosystem function experiments based on randomly selected species assemblages[5, 56].

## Methods

**Study site.** The study area, located in Kilpisjärvi, NW Finnish Lapland (69.055°N, 20.887°E), has a short growing season (from early June to mid-September) and mean annual temperature of −2.0 °C. Our study site is a species-rich tundra meadow, dominated by grasses and forbs, that is located 750 m a.s.l. (50–150 m above the tree line) and usually snow-free from the beginning of June until October. Kilpisjärvi is an important area for semi-domesticated reindeer (*Rangifer tarandus tarandus* L.) in the summer, which traditionally graze in the area from late June to August when there is enough green vegetation to support grazing. Number of reindeer in the area has varied considerably in recent decades[57] fluctuating between ~800 and 1500 animals (corresponding to a density of ca 9–17 animals per km²; personal communication with local reindeer herders) in our study period. The most abundant small mammalian herbivores in the area are Norwegian lemmings (*Lemmus lemmus* L.) and gray-sided voles (*Clethrionomus rufocanus* Sund.). Mountain hares are encountered occasionally (*Lepus timidus* L.).

**Experimental design.** In August 2009 we established 56 plots (each 0.8 × 0.7 m) at our study site[16, 38], and randomly assigned them (by blindly assigning a tag with

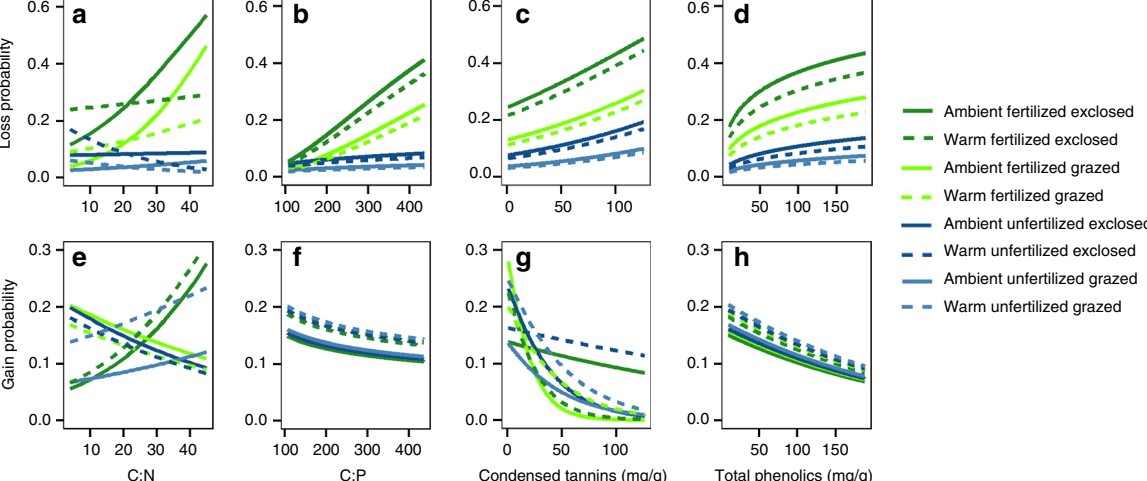

**Fig. 5** Impacts of leaf chemistry traits on probabilities of species losses and gains. Modeled probability of species' losses and gains as functions of carbon: nitrogen ratio (C:N, **a**, **e**), carbon:phosphorus ratio (C:P, **b**, **f**), condensed tannins **c**, **g** and total phenolics **d**, **h** under combinations of warming, grazing and fertilization treatments. Fertilized plots are shown in *green*, unfertilized in *blue*. Herbivore exclosures are indicated with *dark color*, grazed plots in *light color*. *Dashed line* indicates prediction for warmed plots, *solid line* for unwarmed (with ambient temperature). Modeling results can be found in Supplementary Tables 3, 4

unique treatment code for each plot) to the following three treatments in a full-factorial design: mammalian herbivore exclusion, fertilization and warming, or were used as controls, resulting in seven replicates per treatment combination. To exclude mammalian herbivores we surrounded 28 plots individually with 1 m high circular fences (160 cm in diameter), extending 15 cm below ground, made of $10 \times 10$ mm² galvanized mesh. The fenced plots were interspersed among unfenced plots. This treatment was designed to enable assessment of general effects of mammalian herbivory in tundra areas and mimic possible scenarios in which reindeer and other mammals may be locally absent or present at very low densities for a limited time[16]. The fertilization treatment involved application (to 28 plots) of fast-dissolving NPK fertilizer (16-9-22) mixed with 1 liter of water twice per growing season (mid-June and end of July), resulting in addition of 9.6 g N, 5.4 g P and 13.2 g K m⁻² per fertilized plot annually[16, 38]. Our nutrient addition treatment was designed to test the importance of nutrient limitation of plant growth and the role of multiple limiting factors[36, 37] in general, and mimic variation in soil nutrient concentrations between habitats of low vs. high nutrient availability and between tundra areas of low vs. high anthropogenic nutrient enrichment[58]. Also, climate warming may accelerate decomposition of soil organic matter and thereby increase nutrient availability in tundra in future[35], although probably to lesser extent than simulated by this fertilization treatment. For warming, we used International Tundra Experiment (ITEX) hexagonal Open Top Chambers (hereafter OTCs) with a maximum basal diameter of 146 cm[38]. While voles and lemmings can move in and out of these chambers (personal observation), OTCs can at least partly prevent reindeer grazing[59]. To enable equal access for all mammalian herbivores to all grazed (unfenced) plots, we removed the OTCs during the 1-month period (July), when reindeer were normally present at our study site (although solitary reindeer may have been present at other times). To avoid differences in reindeer accessibility between OTC plots and control plots, we erected a temporary reindeer fence around the whole experimental area from the beginning of August to the beginning of July in the following year. This ensured that grazing pressure was similar in all plots, but could have resulted in a small underestimation of the effects of reindeer. Our warming treatment therefore simulated spring and autumn warming (i.e., April–June and August–October) and corresponds well with predictions that temperature increases will be greater in spring and autumn than in July in our study region[60]. Although our warming treatment should be realistic, due to 1-month break in warming our results regarding warming is probably conservative and may underestimate the potential effects of warming. The OTCs increased air temperature by 1.92 °C on average in June (mean ± s.e.m. in control plots and OTCs: 11.20 ± 0.59 °C, $n = 4$, and 13.12 ± 0.25, $n = 4$ °C, respectively) and by 1.23 °C in August 2011 (mean ± s.e.m. in control plots and OTCs: 9.68 ± 0.21 °C, $n = 4$, and 10.91 ± 0.49 °C, $n = 2$, respectively). Moreover, they increased growing degree-days (base daily mean temperature, + 5 °C) on average by 20% in June (2012–2014) and 11% in August and September (2012–2013).

**Measures on plants**. We recorded species occurrence and abundance at peak biomass between 26 July and 5 August before experimental treatments in 2009 and again in 2014 using a modified point intercept method[61] with 108 systematically distributed points in 25 × 50 cm² quadrats. At each point, a pin was lowered to the ground, and the number of hits of each vascular plant species with the pin was recorded. One hit per pin was recorded for bryophyte species, since they grow

virtually in a single layer. The total number of hits per species in each plot was standardized to hits per 100 pins, which was then used as an index of abundance for vascular plants and bryophytes. Results of this non-destructive method, which is widely used to estimate abundance[42, 43, 62], correlate well with true biomass[61]. Certain species were grouped (Supplementary Table 5) as identification of sterile individuals was not possible. Nomenclature for vascular plants follows Hämet-Ahti 1998[63]. We measured the following six functional traits for all vascular species and dominant species of each group of taxa (Supplementary Table 5): height (cm), specific leaf area (SLA; mm² g⁻¹ dry mass), foliar carbon to nitrogen ratio (C:N; based on % C and N), carbon to phosphorus ratio (C:P; based on % C and P), condensed tannins (mg g⁻¹) and total phenolics (mg g⁻¹) following standard protocols[64]. Trait data collected from 10 naturally occurring individuals per species in the study region in summers of 2010 and 2014 were complemented with information regarding traits of four species from the TRY database[65] (Supplementary Table 5). For nutrient analysis we collected two grams of dry leaf material. Nitrogen and carbon were analyzed by combustion in automated elemental analyzer EA1110 CHNS-O (CE instruments). Phosphorus was analyzed color-imetrically after ascorbic acid digestion[66]. Condensed tannins and total phenolics were extracted in 50% acetone. Tannins were then analyzed by a modified version of the vanillin method[67] (the results are expressed as catechin equivalents) and phenolics using the Price and Buttler method[68] (results are expressed as tannic acid equivalents). Species with tannin concentrations under the detection limit (2 mg g⁻¹) were assigned a value of 1 mg g⁻¹. Light availability was measured with an AccuPAR LP-80 Ceptometer (Decacon Devices) above the canopy and in three locations at ground level in each plot on 25 August 2013. We calculated light availability as the proportion of mean ambient photosynthetically active radiation (PAR) at ground level.

**Data analysis**. We calculated changes in species richness for all plant species, vascular plants and bryophytes as the number of species per plot in 2014 minus the number in 2009. The effects of warming, grazing, fertilization and their interactions on changes in species richness of these three plant groups were then analyzed using three separate three-way ANOVAs. We also calculated numbers of lost and gained total plant species, vascular plants and bryophytes between 2009 and 2014, and used five separate three-way ANOVA models, similar to those mentioned above, to test treatment effects on these taxa. Numbers of gained bryophytes were virtually zero and therefore not tested. We used logistic regression models with binomial response variables and adaptive Gauss-Hermite Quadrature estimation to analyze effects of tested factors on species' loss and gain probabilities. For these models, we created two datasets, one compiling species losses and the other compiling species gains. The species loss dataset contained records (the presence/absence in 2009 and 2014, abundance index in 2009 and trait data) for all vascular species in each plot that were present in 2009 (thus species that could potentially be lost; 1000 observations of 43 vascular plant species). The species gain dataset contained records for all vascular species in every plot that were absent in 2009 and recorded in at least one of the plots in 2014 (thus species gained in at least one of the plots; 1257 observations of 37 species). If a species was observed on a plot in 2009, it could not be gained, and was thus not included in the gain dataset. First, we tested whether species' initial abundance (i.e., rarity, measured as number of hits per 100 pins), light availability and their interaction affected species loss probability.

Next, we tested whether functional traits explained species loss likelihood under different treatment combinations, after taking species' initial abundance into account. Effects of each trait were analyzed in a separate model, resulting in six separate trait-models. Fixed explanatory variables in these models were species' initial abundance, the focal trait, warming, herbivore exclusion and fertilization. We allowed these models to include all 4-way and lower interactions, but were not interested in interactions between initial abundance and traits. Finally, we tested whether functional traits affected likelihoods of species gains by using a focal trait, warming, herbivore exclusion, fertilization and their interactions as fixed variables in logistic regression models. This also resulted in six separate models for gain probabilities. All models included plot as a nested random variable. Fixed variables were log-transformed if models failed to converge; transformed factors are indicated in Supplementary Tables 3 and 4. We simplified the logistic regression models by removing non-significant interactions if deletion did not affect model fit, according to a likelihood ratio test ($\chi^2$, $P > 0.1$). The Lme4 package[69] was used for logistic regression models, and all analyses were run in R (R core team 2014, version 3.1.2).

**Data availability**. Data analyzed in this study and the R scripts are available from the corresponding author on reasonable request.

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

## Acknowledgements

We are grateful to Eric Post and Harry Olde Venterink for their comments that improved the manuscript. Konsta Happonen, Suvi and Sini Katves, Nunnu Raatikainen, and Elisa Matikainen are thanked for valuable help in the field. Kilpisjärvi Biological Station provided accommodation and laboratory facilities during the field work. This study was funded by grants from the JC Kempe Memorial Fund, Societas pro Fauna et Flora Fennica, Oskar Öflunds Stiftelse and Swedish Research Council (2015-00498) to E.K., from the Academy of Finland (projects 253385 and 297191) to A.E. and from the Nordic Centre of Excellence—Tundra and the Swedish Research Council for Environment, Agricultural Sciences and Spatial Planning (2006–1539) to J.O. The study has been supported by the TRY initiative on plant traits (http://www.try-db.org)[65]. The TRY initiative and database is hosted, developed and maintained by J.Kattge and G.Bönisch (Max Planck Institute for Biogeochemistry, Jena, Germany). TRY is currently supported by DIVERSITAS/Future Earth and the German Centre for Integrative Biodiversity Research (iDiv) Halle-Jena-Leipzig.

## Author contributions

E.K., A.E. and J.O. designed the experiment. E.K. and A.E. initiated the field experiment. E.K. maintained the field experiment, conducted plant community surveys, analyzed the data and wrote the first draft of the manuscript, to which all authors substantially contributed.

## Additional information

**Competing interests:** The authors declare no competing financial interests.

**Change history:** A correction to this article has been published and is linked from the HTML version of this paper.

