## [Peer Review File · Nature Communications]

REVIEWERS' COMMENTS:

Reviewer #1 (Remarks to the Author):

This is a revised manuscript, reporting on a study examining how tundra plant composition and richness responds to (factorial treatments) of warming, the exclusion of grazers, and fertilizer addition. Notable results include that herbivory can compensate for warming-associated declines in diversity; that high species losses with herbivore exclusion and fertilization are related biomass gains (and decreased light); and that species traits related to resource acquisition (e.g. plant height) can explain species losses and gains (which the study is able to disentangle).

I had several concerns with the original submission, related to the novelty of the study (which wasn't well described), justification for the fertilization treatment (also not well described), and a few details on the study design (which weren't clear). In my opinion, the authors have done a great job addressing all these concerns with a substantially revised manuscript, which addresses all of my main concerns.

I do have additional suggestions that could to increase the clarity of the manuscript. I list these below with line numbers. Most of these are editorial, and the authors should feel free to take these suggestions or not, as they wish. I do suggest the authors think carefully about figures illustrating loss / gain results; specifically, showing both for all traits (they only show a subset in the manuscript and supplemental), and considering carefully which to show in the manuscript vs. the supplementary information. Since this study is able to address both losses and gains (which not all studies are), and this is the main focus of the manuscript, I think the authors should consider showing parallel loss / gain figures in the main text for some of the traits.

- Lines 20 and 22 (abstract), the word 'the' is missing (i.e. 'the diversity of communities' and 'the effects of warming').
- Line 26 (abstract), replace 'it' with 'warming' (i.e. 'while in the absence of herbivores warming caused...')
- Lines 28-30 (abstract), in my opinion, one of the notable aspects of this manuscript is that it demonstrates that multiple global change factors interact in complex ways to influence diversity. It would be possible to allude to this in the last sentence without adding much text, for example by adding the phrase 'and eutrophication' to the end of the sentence.
- Line 93, end of the sentence is missing a word after 'the'; perhaps 'global change factors' (e.g. 'and which traits predict species' susceptibility to these global change factors').
- Line 95, could add the phrase 'including the Arctic' after 'which in many areas' (e.g. 'which in many areas, including the Arctic, are likely to...')
- Line 115, at this point, the reader does not know what 'E' and 'W' represent (in the phrase 'W x E interaction'). I think most will understand that W refers to warming, but E for herbivore exclusion might not be as obvious to everyone. I suggest defining these shortcuts the first time they are mentioned, since the methods come after the results / discussion.
- Lines 131-135, I found these last three sentences hard to follow, and it also seemed a bit clumsy with two uses of the phrase 'In contrast' in the latter half of this paragraph. How about the following (assuming this captures the intended meaning) 'However, warming-induced losses were only observed when herbivores were excluded (2 species lost per plot), and these losses were even greater when plots were additionally fertilized (five species lost per plot). Since no species were lost in plots that were only warmed, these results highlight the synergistic effects of global change, with herbivores playing a particularly important role in influencing diversity.'
- Line 138, I suggest removing 'Moreover' or 'most strikingly' from the beginning of this sentence (e.g. 'Moreover, our results...' or 'Strikingly, our results'), having both makes the sentence

unnecessarily clumsy.

- Line 145: remove the word 'only' from this sentence.
- Lines 146-149, since the Post study is called out specifically, and has contrasting results to this study, it might be good to mention in this paragraph that this suggests more work needs to be done on this topic.
- Line 155, insert 'the' before reduction (i.e. 'amplified the reduction of species richness').
- Line 159, is there a figure or table to support the statement that light / biomass related impacts on diversity were solely mediated by impacts on species loss, not gains? It would be nice to point the reader to a part of the manuscript or supplemental so they can verify this result themselves. How about showing the same relationships (species gained vs. light) in a supplemental figure?
- Line 183-184, what does the sentence 'Our experimental treatments dictated the relative importance of traits for loss likelihood'? Should dictated be replaced by 'revealed' (i.e. 'Our experimental treatments revealed the importance of some traits for species loss')?
- Lines 184-186, this sentence is confusing because warming did cause changes in richness (albeit through impacts on species gains). How about changing the sentence to 'Although warming clearly influenced richness, it did so by influencing gains, not losses. Thus, we were unable to identify traits that were predictive of species that are disproportionately lost from plots under warming.'
- Lines 191-197, Fig. 4a should be cited earlier in this section.
- Lines 217-218, I recommend changing this sentence to 'Although warming, either singly or in combination with other treatments, did not result in species losses, it did increase species gains in unfertilized plots with herbivores.'
- Lines 240-243, quite a long sentence, and it also misses the opportunity to mention eutrophication. How about replacing the one long sentence with two sentences along the lines of 'Our results suggest that if climate warming coincides with losses of mammalian herbivores (such as reindeer or lemmings) in tundra, major decreases in diversity are likely as a consequence of losses of short-statured, resource-conservative species. Our results also reveal that such species losses are likely to be even greater in more productive areas, or where N-deposition is high.'
- Lines 252-327, methods – these are much clearer and easier to follow.
- Lines 362-364, what does 'reasonable request' mean? Why not just post the data in a data repository (e.g. Dryad)?
- Lines 540-552 (Fig. 3 & 4), It would be nice for visual comparison to have all three panels of each graph next to each other; perhaps with the legend below.
- I wonder if the authors might consider showing both loss and gain probability as a function of traits in one of the figures in the main text (rather than relegating all of these to the supplementary). The second reviewer pointed out that one of the important contributions of this study is the ability to parse out changes in diversity due to losses and gains. This is something I concur with (although I had not picked up on this in my original review). For example, the species gain vs. height shows interesting patterns, as does SLA. It's clear that not all should probably be shown for length, but showing a few critical ones would be interesting, especially since a side by side comparison indicates that they often, but not always, show opposite patterns (SLA vs. height). Moreover, they can be added to existing figures (e.g. a 2x3 panel of figures, as opposed to a 1x3).
- Supplement Fig. 3, where are the plots of species gain relative to C:P and C:N? The effects of height, C:P and C:N on species loss are presented in the main text, but only height vs. gain is presented here. I understand from the text that there is no significant relationship, but showing that with a figure would actually be informative. Following from the point made above, it would also be good to show all.

Reviewer #2 (Remarks to the Author):

The authors did an excellent job reframing this story. The novelty now jumps out at me, and I commend their great effort in doing so. I believe this will make the manuscript much more highly cited. They have addressed all my concerns. Below I list several minor things which I hope will improve the manuscript. Overall, this is great manuscript, and I thoroughly enjoyed reading your rewrite.

Detailed comments:

Abstract – First two sentences are just a little bit boring. The rest of the abstract is wonderful.

Main Text

Line 182 – Introduce random-loss hypothesis sooner – somewhere in the intro. Random loss to me would mean that species are lost randomly irrespective of starting biomass. But that does not appear to be what you mean here. In my experience, non-random loss is that rare species are lost from the community first which is what you see except for when light becomes severely limited.

Line 217 - clunky

Responses to reviewer comments

Reviewer #1 (Remarks to the Author):

This is a revised manuscript, reporting on a study examining how tundra plant composition and richness responds to (factorial treatments) of warming, the exclusion of grazers, and fertilizer addition. Notable results include that herbivory can compensate for warming-associated declines in diversity; that high species losses with herbivore exclusion and fertilization are related biomass gains (and decreased light); and that species traits related to resource acquisition (e.g. plant height) can explain species losses and gains (which the study is able to disentangle).

I had several concerns with the original submission, related to the novelty of the study (which wasn't well described), justification for the fertilization treatment (also not well described), and a few details on the study design (which weren't clear). In my opinion, the authors have done a great job addressing all these concerns with a substantially revised manuscript, which addresses all of my main concerns.

Thank you very much for these kind words.

I do have additional suggestions that could to increase the clarity of the manuscript. I list these below with line numbers. Most of these are editorial, and the authors should feel free to take these suggestions or not, as they wish. I do suggest the authors think carefully about figures illustrating loss / gain results; specifically, showing both for all traits (they only show a subset in the manuscript and supplemental), and considering carefully which to show in the manuscript vs. the supplementary information. Since this study is able to address both losses and gains (which not all studies are), and this is the main focus of the manuscript, I think the authors should consider showing parallel loss / gain figures in the main text for some of the traits.

- Lines 20 and 22 (abstract), the word 'the' is missing (i.e. 'the diversity of communities' and 'the effects of warming').

Corrected.

- Line 26 (abstract), replace 'it' with 'warming' (i.e. 'while in the absence of herbivores warming caused...')

Replaced.

- Lines 28-30 (abstract), in my opinion, one of the notable aspects of this manuscript is that it demonstrates that multiple global change factors interact in complex ways to influence diversity. It would be possible to allude to this in the last sentence without adding much text, for example by adding the phrase 'and eutrophication' to the end of the sentence.

Great suggestion, added.

- Line 93, end of the sentence is missing a word after 'the'; perhaps 'global change factors' (e.g. 'and which traits predict species' susceptibility to these global change factors').

Reworded.

- Line 95, could add the phrase 'including the Arctic' after 'which in many areas' (e.g. 'which in many areas, including the Arctic, are likely to...')

We added 'including tundra'.

- Line 115, at this point, the reader does not know what 'E' and 'W' represent (in the phrase 'W x E interaction'). I think most will understand that W refers to warming, but E for herbivore exclusion might not be as obvious to everyone. I suggest defining these shortcuts the first time they are mentioned, since the methods come after the results / discussion.

Clarified.

- Lines 131-135, I found these last three sentences hard to follow, and it also seemed a bit clumsy with two uses of the phrase 'In contrast' in the latter half of this paragraph. How about the following (assuming this captures the intended meaning) 'However, warming-induced losses were only observed when herbivores were excluded (2 species lost per plot), and these losses were even greater when plots were additionally fertilized (five species lost per plot). Since no species were lost in plots that were only warmed, these results highlight the synergistic effects of global change, with herbivores playing a particularly important role in influencing diversity.'

We modified this sentence.

- Line 138, I suggest removing 'Moreover' or 'most strikingly' from the beginning of this sentence (e.g. 'Moreover, our results...' or 'Strikingly, our results'), having both makes the sentence unnecessarily clumsy.

'Moreover' removed.

- Line 145: remove the word 'only' from this sentence.

Removed.

- Lines 146-149, since the Post study is called out specifically, and has contrasting results to this study, it might be good to mention in this paragraph that this suggests more work needs to be done on this topic.

Good suggestion, a notion added.

- Line 155, insert 'the' before reduction (i.e. 'amplified the reduction of species richness').

Added.

- Line 159, is there a figure or table to support the statement that light / biomass related impacts on diversity were solely mediated by impacts on species loss, not gains? It would be nice to point the reader to a part of the manuscript or supplemental so they can verify this result themselves. How about showing the same relationships (species gained vs. light) in a supplemental figure?

We have now referred to Figures 1b and Supplementary Fig. 1, which both show that three-way manipulation did not change number of gained species. We also added Supplementary Fig 2 showing that light availability had not impact on probability of species gains.

- Line 183-184, what does the sentence ‘Our experimental treatments dictated the relative importance of traits for loss likelihood’? Should dictated be replaced by ‘revealed’ (i.e. ‘Our experimental treatments revealed the importance of some traits for species loss’)?

Thank you, reworded as suggested.

- Lines 184-186, this sentence is confusing because warming did cause changes in richness (albeit through impacts on species gains). How about changing the sentence to ‘Although warming clearly influenced richness, it did so by influencing gains, not losses. Thus, we were unable to identify traits that were predictive of species that are disproportionately lost from plots under warming.’

We clarified these sentences.

- Lines 191-197, Fig. 4a should be cited earlier in this section.

We now cite the new trait figures (Fig. 4-5) earlier in the section.

- Lines 217-218, I recommend changing this sentence to ‘Although warming, either singly or in combination with other treatments, did not result in species losses, it did increase species gains in unfertilized plots with herbivores.’

Changed as suggested.

- Lines 240-243, quite a long sentence, and it also misses the opportunity to mention eutrophication. How about replacing the one long sentence with two sentences along the lines of ‘Our results suggest that if climate warming coincides with losses of mammalian herbivores (such as reindeer or lemmings) in tundra, major decreases in diversity are likely as a consequence of losses of short-statured, resource-conservative species. Our results also reveal that such species losses are likely to be even greater in more productive areas, or where N-deposition is high.’

Thank you for this great suggestion, we added the suggested sentences.

- Lines 252-327, methods – these are much clearer and easier to follow.

Thank you very much!

- Lines 362-364, what does 'reasonable request' mean? Why not just post the data in a data repository (e.g. Dryad)?

We do understand the importance of data sharing for open science and facilitating scientific process. However, we are currently using the same dataset for answering other research questions, and will post data in Dryad once we have finished these manuscripts.

- Lines 540-552 (Fig. 3 & 4), It would be nice for visual comparison to have all three panels of each graph next to each other; perhaps with the legend below.

We changed the panels of Fig. 3 next to each other. Figure 4 has been changed (see below.)

- I wonder if the authors might consider showing both loss and gain probability as a function of traits in one of the figures in the main text (rather than relegating all of these to the supplementary). The second reviewer pointed out that one of the important contributions of this study is the ability to parse out changes in diversity due to losses and gains. This is something I concur with (although I had not picked up on this in my original review). For example, the species gain vs. height shows interesting patterns, as does SLA. It's clear that not all should probably be shown for length, but showing a few critical ones would be interesting, especially since a side by side comparison indicates that they often, but not always, show opposite patterns (SLA vs. height). Moreover, they can be added to existing figures (e.g. a 2x3 panel of figures, as opposed to a 1x3).

We now present loss and gain probability as a function of traits in two figures in the main text: Fig. 4 contains 2x2 panels and shows height and SLA and Fig. 5 contains 2x4 panels showing all leaf chemistry traits.

- Supplement Fig. 3, where are the plots of species gain relative to C:P and C:N? The effects of height, C:P and C:N on species loss are presented in the main text, but only height vs. gain is presented here. I understand from the text that there is no significant relationship, but showing that with a figure would actually be informative. Following from the point made above, it would also be good to show all.

All these panels are now shown in figures 4-5.

Reviewer #2 (Remarks to the Author):

The authors did an excellent job reframing this story. The novelty now jumps out at me, and I commend their great effort in doing so. I believe this will make the manuscript much more highly cited. They have addressed all my concerns. Below I list several minor things which I hope will improve the manuscript. Overall, this is great manuscript, and I thoroughly enjoyed reading your rewrite.

We are very grateful for these comments, thank you!

Detailed comments:

Abstract – First two sentences are just a little bit boring. The rest of the abstract is wonderful.

These sentences have now been modified.

Main Text

Line 182 – Introduce random-loss hypothesis sooner – somewhere in the intro. Random loss to me would mean that species are lost randomly irrespective of starting biomass. But that does not appear to be what you mean here. In my experience, non-random loss is that rare species are lost from the community first which is what you see except for when light becomes severely limited.

Thank you for taking up this point. Since random loss can be defined differently by different authors (see eg. Suding et al 2005, Smith & Knapp 2003, Isbell *et al.* 2013), we removed random-loss hypothesis and speak now about rarity and traits influencing loss probability.

Line 217 - clunky

This is now reworded as suggested by reviewer 1.

Isbell, F. *et al.* Nutrient enrichment, biodiversity loss, and consequent declines in ecosystem productivity. *Proc. Natl. Acad. Sci.* **110**, 11911–11916 (2013).

Smith M. D. & Knapp A. K. Dominant species maintain ecosystem function with non-random species loss. *Ecol. Lett.* **6**, 509–517 (2003).

Suding, K. N. *et al.* Functional- and abundance-based mechanisms explain diversity loss due to N fertilization. *Proc. Natl. Acad. Sci. U. S. A.* **102**, 4387–4392 (2005).